# Optimization and Spatiotemporal Differentiation of Carbon Emission Rights Allocation in the Power Industry in the Yangtze River Economic Belt

**Dalai Ma, Yaping Xiao ***  **and Na Zhao**

School of Management, Chongqing University of Technology, Chongqing 400054, China;
madalai@cqut.edu.cn (D.M.); zhaona1997@2020.cqut.edu.cn (N.Z.)
* Correspondence: ashley_air@163.com

**Abstract:** Reasonable allocation of carbon emission rights aids in the realization of the goal of carbon emission reduction. The purpose of this paper is to examine how carbon emission rights in the power sector in the Yangtze River Economic Belt (the YREB) are distributed. The YREB spans China's eastern, central, and western areas. The levels of development and resource endowment differ significantly across regions, resulting in great heterogeneity in the YREB provinces' carbon emission rights distribution in the power sector. The ZSG–DEA model is used in this paper to re-adjust the power sector's carbon emission quotas in each province to achieve optimal efficiency under the country's overall carbon emission reduction target. The results show that: (1) In most provinces, the power sector's initial distribution efficiency is inefficient. Only Zhejiang and Yunnan have reached the production frontier, with Jiangxi and Chongqing having the lowest distribution efficiency. In the future, we should concentrate our efforts on them for conserving energy and lowering emissions; (2) The initial distribution efficiency of the power sector in the YREB's upstream, midstream, and downstream regions is considerably different. Most upstream and downstream provinces have higher carbon emission quotas, while most midstream provinces have less, implying that the power sector in the midstream provinces faces greater emission reduction challenges; (3) The carbon emission quotas of the power industry varies greatly between provinces and shows different spatial features over time. In the early stage (2021–2027), the carbon emission quota varies substantially, while for the later stage (2027–2030), it is rather balanced. Zhejiang, Jiangsu, Sichuan, and Yunnan are more likely to turn into sellers in the market for carbon emission trading with larger carbon emission quotas. While Jiangxi and Chongqing are more likely to turn into buyers in the market for carbon emission trading with fewer carbon emission quotas. Other provinces' carbon emission quotas are more evenly distributed. To successfully achieve China's emission reduction target by 2030, the YREB should promote regional collaboration, optimize industrial structure, accelerate technical innovation, establish emission reduction regulations, and provide financial support based on local conditions.

**Keywords:** carbon emission rights allocation; ZSG-DEA; China's power sector; YREB; spatiotemporal differentiation

## 1. Introduction

Since the reform and opening up of China, its economy has been rapidly expanding and has grown to become the world's second largest (Ma and Cai) [1]. As a result, massive amounts of energy, especially fossil fuels, are consumed. The source of most carbon emissions is fossil fuels. In 2017, carbon emissions in China were responsible for 28 percent of total global emissions, far outnumbering the second-largest emitter, the United States (15%) (BP2018) [2]. China presented the dual carbon goal of "striving to peak carbon dioxide emissions by 2030 and achieve carbon neutrality by 2060" at the 75th United Nations General Assembly to promote global decarbonization, which not only brought huge pressure on achieving carbon emission reductions to China, but also made higher requirements for

China's power industry to develop in a low-carbon manner. As a pillar industry in China (Zhou et al.) [3], the power industry needs to meet the demand for electricity generated by daily economic development, industrial production, and urbanization. However, China's current power generation mode is primarily thermal (Li et al.) [4], and the amount of power generated by clean power generation methods is far from meeting society's needs, resulting in a large amount of $CO_2$. Nearly 40% of China's carbon emissions are attributed to the power sector (Yu et al.) [5], which has put enormous pressure on energy conservation and environmental protection. Therefore, reducing carbon emissions in the power sector is critical to achieve China's goal of reducing carbon emissions. Different regions and industries have different emission reduction targets under the overall goal of "dual carbon" because different regions have varying levels of technological development and are at various stages of development. To achieve maximum efficiency, the ZSG-DEA model can adjust each province's carbon emission allowances in accordance with the country's overall carbon reduction targets.

The YREB connects 11 Chinese provinces, with a population and economic aggregate that exceeds 40% of the country's total (Li et al.) [6], having a significant influence on China's overall social and economic development. However, the YREB has faced severe resource and environmental problems because of long-term high-intensity industrial economic development, particularly the increasingly serious carbon emission problem. The YREB is rich in hydropower and mineral resources. It is an important hydropower and pithead thermal power supply area in China, as well as a major transmission source for "West-East Power Transmission". According to statistics data [7], its power generation accounts for 37.8 percent of China's total, producing a significant amount of carbon emissions, making it a key area for reduction of carbon emissions. The YREB connects China's eastern, central, and western regions. The level of industrial development and resource endowment varies greatly across these regions. However, does the allocation of carbon emission rights in the YREB's power sector also show significant differences? This is an important issue that needs to be addressed.

The remainder of this paper is laid out as follows. Section 2 examines the relevant studies on carbon emission rights allocation. The research methods and data are described in Section 3. And the results of the study are listed in Section 4. Finally, Section 5 summarizes the research and presents implications for policy.

## 2. Literature Review

Low-carbon development has grown in importance as a research topic in academic circles of various countries in the recent past. The allocation of carbon emissions has also attracted increasing attention from scholars. The main point of contention in existing research is the selection of principles and methods for allocating carbon emissions rights. Most scholars recognize the principles of fairness and efficiency at the level of distribution. The United Nations Framework Convention on Climate Change (UNFCCC) [8] identified the principle of fairness and proposed "shared but distinct responsibilities" in addressing changes in the climate. Fairness, as a relatively broad concept, encompasses not only egalitarianism and historical emission responsibility, but also the ability to reduce emissions. As a result, when allocating carbon emission rights, scholars typically consider population, historical carbon emissions, and economic level. Pan et al. [9] proposed a distribution scheme based on per capita cumulative emissions to create a global carbon emission space that is fair. Zhu et al. [10] proposed that the development performance of various industries be considered to reflect the fairness of carbon emission rights allocation. In addition, scholars frequently use multiple indicators to allocate carbon emissions, because the principle of fairness necessitates the use of multiple indicators. To simulate carbon allowance allocation in the Beijing-Tianjin-Hebei region, Han et al. [11] created a comprehensive index and used a comprehensive weighting method: GDP per capita, cumulative $CO_2$ emissions, and energy consumption per unit of industrial added value were chosen to represent carbon emission reduction capability, potential, and responsibility. Fang et al. [12] discussed the

optimal allocation of carbon emission rights based on energy equity, as well as the method for optimizing the allocation scheme under GDP constraints, population, fossil energy, and ecological production land. According to their findings, the importance of fossil energy resources and ecological production land was greater. Furthermore, as a widely used indicator for evaluating fairness, the Gini coefficient is frequently used to ensure that carbon emission allocation results are equitable (Fang et al. [12], Guo et al. [13], He et al. [14]). Through the above research, it is found that the carbon emission allocation principle is more concerned with absolute fairness and ignores the perspective of efficiency, which is not conducive to a reasonable and effective distribution of carbon emissions.

People are becoming more aware, as research into the distribution of carbon emission rights advances, that the so-called "absolutely fair" distribution of carbon emission rights does not benefit all countries and regions (He and Zhang [14], Kong et al. [15], He et al. [16]). The principle of equity considers differences in low-carbon development levels across provinces, but they were not practical. It cannot effectively motivate provinces with better low-carbon development while restricting provinces with backward low-carbon development, and it cannot improve overall efficiency. Another important principle to consider when allocating carbon emission rights is the efficiency principle of profit maximization (Du et al.) [17], which according to Zhou et al. [18], is the highest economic return for the least amount of resources. Carbon emission rights can help a country's economy grow as a valuable resource, but they are restricted and should be distributed in a scientific and rational manner. Therefore, Zhou [19], Qin et al. [20] and Liu et al. [21] studied the optimal allocation of carbon dioxide emissions by the DEA method, cooperative game model and nonlinear programming method respectively.

Scholars have studied the method of allocating carbon emission rights based on the principles of fairness and efficiency extensively. Methods for allocating carbon emission rights in the past have included the grandfather method (Schmidt and Heizig) [22], the benchmark method (Sartor et al. [23], Zhang et al. [24]), the auction method (Burtraw and McCormack) [25], the indicator method (He and Zhang [14], Zhao et al. [26]) and others. These methods, to some extent, ensure the fairness of the distribution of carbon emission rights among decision-making units, but they ignore another distribution principle—efficiency. They considered differences in low-carbon development levels across provinces, but they were not practical. It cannot effectively motivate provinces with better low-carbon development while restricting provinces with backward low-carbon development, and it cannot improve overall efficiency. In contrast, data envelopment analysis (DEA), an optimization method aimed at improving overall system efficiency, has been introduced into the study of carbon emission rights allocation. Because countries and regions often set carbon emission targets, the total amount of carbon emissions should be limited within a certain range when allocating carbon emission rights. In this situation, how can you achieve maximum efficiency? The zero sum gains DEA (ZSG–DEA) model was proposed by Lins et al. [27] as a viable solution to this problem. It has since become a widely used method for allocating carbon emission rights. Gomes and Lins et al. [28] and Chiu et al. [29] used the ZSG–DEA model to investigate the distribution of carbon emission rights. Furthermore, the ZSG–DEA model has been used by some researchers in China to examine carbon emission allowances at the provincial level (Cai and Ye [30], Yang et al. [31], Cui et al. [32]) and at the industry level (Chen et al. [33]). However, there has been little research done on the distribution of carbon emission rights across a large part of China. Zhuang et al. [34] also mentioned that future research in different geographic clusters in China could be conducted to build a more appropriate carbon dioxide emission allocation mechanism.

In response to the aforementioned issues, this paper achieves breakthroughs in two aspects. Firstly, the power sector is the leading source of carbon emissions, but few academics have been concerned about its issue of carbon emission quotas. Therefore, the allocation of carbon emission rights in the YREB's power industry is the subject of this paper. The analysis of its carbon emission rights allocation can provide more precise information

for the rational allocation of carbon emission quotas in the power sector. Secondly, few studies have focused on the temporal and spatial differentiation of carbon emission rights allocation in the power industry, whereas this paper does. Paying attention to this will aid in resolving the problem of heterogeneity in the distribution of carbon emission rights in the power sector in the provinces of the YREB, as well as assisting provinces in formulating accurate carbon emission reduction targets.

This paper first forecasts the input-output variables of the power industry in the YREB from 2021 to 2030. The ZSG–DEA model is then used to calculate the carbon emission rights distribution efficiency in the YREB power industry, iterate carbon emission allowances, optimize carbon emission allowances, and establish a reasonable allocation scheme. Unlike previous research, which has focused solely on carbon emission allocation in 2030, this paper examines carbon emission allocation in each year. After that, the temporal and spatial evolution characteristics of carbon quotas are analysed to compare differences in carbon emission reduction responsibilities and emission reduction paths among provinces over the last decade. The results can then enable recommendations for achieving the YREB's low-carbon development as well as improvements to the power carbon market trading system.

## 3. Materials and Methods

### 3.1. ZSG–DEA Model

Data Envelope Analysis (DEA) was proposed in 1978 by American operations researcher Charnes et al. [35] and is a widely used method in academia to assess the relative efficiency of homogeneous decision-making units. The conventional DEA model assumes that each decision-making unit's inputs and outputs are independent of one another. When the DEA model is used in the distribution field, however, it is constrained by the requirement that a certain input indicator (or output indicator) keeps the total amount unchanged. The traditional DEA model will fail in this case. Lins and Gomes et al. [28] proposed a zero-sum gains DEA model, which we call the ZSG–DEA model, in response to this problem. The ZSG–DEA model places all decision-making units on a new aim while keeping the sum of the changed variables constant, because the inputs (or outputs) of decision-making units that were previously technically ineffective under traditional DEA have been reconfigured. The allocation of quotas among provinces in China is competitive, based on the premise of a certain amount of carbon emission quotas. In other words, an increase in emissions in some provinces results in a decrease in emissions in others, reflecting the zero-sum gains concept of constant total emissions. When the zero-sum income concept is applied to the power industry, the total carbon emissions of the power industry are limited. Adjust the distribution of power carbon emission rights in all provinces on a regular basis to achieve the optimal distribution, which will also promote more benign economic development.

Since the ZSG–DEA model was proposed, its application in carbon emission rights allocation has been continuously improved. The focus of the debate is on the treatment of input and output values. Scholars have proposed ZSG–DEA models with competitive input (Cui et al. [32], Yang et al. [31], Fang et al. [36]) or competitive output (Zhuang et al. [34]). Because there are, in reality, both competitive and non-competitive inputs and outputs, the ZSG–DEA model considering the goal of maximizing global efficiency was proposed. Based on the model setting of Wu et al. [37], this paper also considers the expansion (or reduction) of non-competitive output or input based on the distribution of output or input with competitive relationship and proposes an improved ZSG–DEA model. Assuming that there are $n$ decision making units (DMU$_j$) (j = 1, . . . , n). Each decision making unit has m competitive input, s non-competitive inputs and q outputs, respectively denoted by $x_{ij}$ (i = 1, . . . , m), $y_{rj}$ (r = 1, . . . , s), and $z_{pj}$ (p = 1, . . . , q). where $\lambda_j$ represents the weight of

DMU$_j$ and the specific decision-making unit is represented by $j_0$. For the ZSG–DEA model with competitive relationships between inputs, it can be expressed as Equation (1).

$$
\min h_{j0}
$$
$$
s.t. \begin{cases}
\sum\limits_{j=1}^{n} \lambda_i x_{ij}\left[1 + \frac{x_{ij0}(1-h_{j0})}{\sum\limits_{j \neq j_0} x_{ij}}\right] \leq h_{j0} x_{ij0} \\
\sum\limits_{j=1}^{n} \lambda_j y_{rj} \leq h_{j0} x_{rj0}, \sum\limits_{j=1}^{n} \lambda_j z_{pj} \geq z_{pj0} \\
\sum\limits_{j=1}^{n} \lambda_j = 1 \\
\lambda_j \geq 0, j = 1, \ldots, n
\end{cases}
\tag{1}
$$

Among them, $h_{j0}$ represents the efficiency value of the DMU$_0$. If DMU$_0$ is an inefficient DEA unit, in order to achieve DEA effectiveness, it must reduce the use of $i$-th input by $u_0 = x_{ij0}(1 - h_{j0})$ and share this amount of input proportionally to other decision-making unit by $\frac{x_{ij0}(1-h_{j0})}{\sum\limits_{j \neq j_0} x_{ij}}$. The quantity obtained by the other decision-making unit is $\frac{x_{ij0}(1-h_{j0})}{\sum\limits_{j \neq j_0} x_{ij}} x_{ij}$. As all DMUs are reducing the proportion of input at the same time, the reallocation of $i$-th input to DMU $_j$ is:

$$
x'_{ij} = \sum_{j \neq j_0} \left[\frac{x_{ij0}(1 - h_{j0})}{\sum\limits_{j \neq j_0} x_{ij}} x_{ij}\right] - x_{ij}(1 - h_j)
\tag{2}
$$

*3.2. Data Source and Processing*

3.2.1. Input-Output Indicators

Labour, capital, and energy consumption are all common inputs in the industrial production function. Output indicators include output value and various industrial pollutants. The power sector of the 11 YREB provinces was the basic decision-making unit in the construction of the model to measure the distribution efficiency of carbon emission rights in China's power industry. This study was based on the indicator settings of Zhou et al. [19] and Zhuang et al. [34], with power labour input, power capital, and power energy as input variables, power output value as the desired output, and power carbon emissions as the undesired output. The differences in resources and economic levels between provinces were considered in these indicators. At the same time, they strived for the highest power output value and the least amount of pollution with the least amount of labour, capital, and energy, reflecting the fairness and efficiency principle of carbon emission rights. The data were derived from the China Statistical Yearbook [7], China Energy Statistical Yearbook [38], and China Provincial Statistical Yearbook. Table 1 shows an explanation for each input-output variable.

**Table 1.** Input-output variables of power carbon emission rights allocation efficiency.

| Variable Classification | Specific Variable | Variable Explanation |
| --- | --- | --- |
| Input variable | Power labour input | Employment of power, thermal and supply sectors |
| | Power capital | Actual capital stock of power, thermal and supply sectors based on 2005 |
| | Power energy | Power consumption |
| Output variable | Power output value | Sales value of power, thermal and supply sectors based on 2005 |
| | Power carbon emissions | Estimated power $CO_2$ emissions by regions |

(1) Power labour force: Employment is often used to represent labour force indicators. Because there are no special statistics on human resource investment in the power industry.

This paper replaced the number of employees in the power industry with the number of employees in the power, thermal and supply sectors, and the data in 2020 was obtained by the moving average method. The average annual population growth rate was calculated for the period 2011–2020. Assuming that the population growth rate remained unchanged from 2021 to 2030, and the proportion of provinces was consistent with 2020, the population of China's power industry from 2021 to 2030 was predicted.

(2) Power capital: This paper, like many previous studies such as Zhuang et al. [34], adopted power capital stock to measure power capital. For the initial capital stock, this paper used the method of Hall et al. [39]. The formula is $K_{i0} = I_{i0}/(\delta + g_i)$, where $I_{i0}$ represents the total fixed capital; $\delta$ represents the depreciation rate, taking 9.6% in this paper; $g_i$ represents the average GDP growth rate in each province. The "perpetual inventory method" was used to calculate the capital stock of each province every year. The calculation formula is $K_{i,t} = I_{i.t} + (1-\delta)K_{i,t-1}$, where $K_{i,t}$ is the capital stock of the $i$-th province during the $t$-th period and $I_{i,t}$ is the investment of $i$-th province in $t$-th period. Then the capital stock should be adjusted to a constant price of 2005. The capital stock for 2021–2030 was predicted by the average growth rate for 2011–2022.

(3) Power energy: Energy was represented by power consumption, and we predicted power consumption from 2021 to 2030 from the average growth rate.

(4) Power output value: Power output value is the "good" output brought by the power production process. The expected output in this study is the industrial sales output value of the power industry in each province after deflator, with 2005 as the base period. In addition, the output data from 2021 to 2030 was forecast based on the power industry's average output growth rate from 2011 to 2020. Due to the lack of special statistics on the output value of the power industry, this paper replaced the output value of the power industry with the sales value of power, thermal and supply, and the data from 2017 to 2020 was obtained by using the moving average method.

(5) Power CO$_2$ emissions: This paper focused on the distribution of carbon emission rights, so carbon emission rights were included as an undesirable output in the distribution efficiency model. The main methods of the DEA model in dealing with undesirable output include undesirable output as an input method, hyperbolic method, reciprocal conversion method, conversion vector method, directional distance function method and SBM model method, etc. Other methods may be confronted with the problem of ineffective solutions, so this paper adopted the CCR model with undesired output as an input to deal with the issue of carbon emissions. The calculation methods of CO$_2$ emissions and energy consumption of each province in China over the years are as in Equation (3). This paper employed the reference method based on terminal consumption in the energy balance table of various regions, which was listed in the *2007 IPCC Guideline on National GHG Inventories* (IPCC, 2007) [40]. Each energy type was calculated based on their individual carbon dioxide emission coefficients, which eliminates the calculation error caused by rough classification.

$$\text{CO}_{2i} = \sum_j E_{ij} \times EFj \times O_j \tag{3}$$

where CO$_{2i}$ epresents the total CO$_2$ emissions from the $i$-th province in Mt (100 million tons); $E_{ij}$ is the physical consumption of the $j$-th energy in the $i$-th province, measured in tons (t) or cubic meters (M$^3$); $EF_j$ denotes the carbon emission coefficient of the $j$-th energy, expressed in t CO$_2$/t or t CO$_2$/M$^3$. The coefficient of the $j$-th energy converted into standard coal is represented by $O_j$. Tables 2 and 3 show the carbon emission coefficients of various energy sources as well as the reference coefficients of standard coal. The data came from the China Statistical Yearbook [7].

**Table 2.** Carbon emission coefficients of various energy sources (t carbon/t standard coal).

| Energy Type | Raw Coal | Coke | Crude Oil | Fuel Oil | Gasoline | Kerosene | Diesel Fuel | Natural Gas | Electricity |
|---|---|---|---|---|---|---|---|---|---|
| Carbon emission coefficients | 0.7476 | 0.1128 | 0.5854 | 0.6176 | 0.5532 | 0.3416 | 0.5913 | 0.448 | 2.2132 |

**Table 3.** Reference coefficients of standard coal for various energy sources.

| Energy Type | Raw Coal | Coke | Crude | Fuel Oil | Gasoline | Kerosene | Diesel Fuel | Natural Gas | Electricity |
|---|---|---|---|---|---|---|---|---|---|
| Standard coal coefficient | 0.7143 tce/t | 0.9714 tce/t | 1.4286 tce/t | 1.4286 tce/t | 1.4714 tce/t | 1.4714 tce/t | 1.4571 tce/t | 13.30 tce/$10^4$ m$^3$ | 1.229 tce/$10^4$ kwh |

### 3.2.2. Calculation of Initial Carbon Emissions Allowance

Because of the large differences in economic performance, natural resources, and historical carbon emissions among provinces, focusing solely on distribution efficiency will result in an imbalance of provinces' carbon emission reduction responsibilities. As a result, we used historical cumulative carbon emissions as the initial distribution standard to ensure the fairness of carbon emission right distribution. The exact calculation procedure was as follows.

To begin, national total carbon emissions and GDP were used to calculate the carbon emission intensity per unit of GDP from 2011 to 2020. Second, from 2021 to 2030, the carbon emission intensity target value was calculated using the goal of "reducing national carbon emission intensity by 65 percent (compared to 2005) by 2030". Finally, the historical cumulative proportion of carbon emissions from 2011 to 2020 in each province was used as the basis for the allocation of carbon emission rights from 2021 to 2030 in the power industry.

### 4. Results and Discussion

Based on Equation (1), we used DEA to calculate the initial value of the power sector in each province in the YREB from 2021 to 2030. Table 4 shows the initial efficiency of carbon emission rights allocation. The initial allocation efficiency of carbon emission rights in each province was low, as shown in Table 4, and there were significant differences between provinces.

**Table 4.** Efficiency of carbon emission rights allocation in the power sector in the YREB from 2021 to 2030.

| Province | 2021 | 2022 | 2023 | 2024 | 2025 | 2026 | 2027 | 2028 | 2029 | 2030 |
|---|---|---|---|---|---|---|---|---|---|---|
| Shanghai | 0.7829 | 0.7675 | 0.7519 | 0.7783 | 0.7803 | 0.7794 | 0.6917 | 0.6879 | 0.6869 | 0.6841 |
| Jiangsu | 0.8605 | 0.8571 | 0.8515 | 0.8401 | 0.8497 | 0.8576 | 0.8639 | 0.8687 | 0.8718 | 0.8734 |
| Zhejiang | 1.0000 | 1.0000 | 1.0000 | 1.0000 | 1.0000 | 1.0000 | 1.0000 | 1.0000 | 1.0000 | 1.0000 |
| Anhui | 0.8317 | 0.7960 | 0.7643 | 0.7332 | 0.7028 | 0.6733 | 0.6446 | 0.6168 | 0.5898 | 0.5638 |
| Jiangxi | 0.4840 | 0.4464 | 0.4117 | 0.3796 | 0.3499 | 0.3251 | 0.3032 | 0.2834 | 0.2654 | 0.2489 |
| Hubei | 0.7435 | 0.7355 | 0.7042 | 0.6685 | 0.6346 | 0.6024 | 0.5719 | 0.5429 | 0.5154 | 0.4892 |
| Hunan | 0.7276 | 0.7242 | 0.7034 | 0.6783 | 0.6540 | 0.6307 | 0.6082 | 0.5864 | 0.6333 | 0.5453 |
| Chongqing | 0.6788 | 0.6674 | 0.6523 | 0.6382 | 0.6258 | 0.6150 | 0.6071 | 0.6002 | 0.5940 | 0.5886 |
| Sichuan | 0.8422 | 0.8097 | 0.7784 | 0.7482 | 0.7191 | 0.6707 | 0.6640 | 0.6379 | 0.6128 | 0.5887 |
| Guizhou | 0.7942 | 0.7819 | 0.7548 | 0.7256 | 0.6976 | 0.6976 | 0.6448 | 0.6199 | 0.5960 | 0.5730 |
| Yunnan | 1.0000 | 1.0000 | 1.0000 | 1.0000 | 1.0000 | 1.0000 | 1.0000 | 1.0000 | 1.0000 | 1.0000 |

Figure 1 depicts the agglomeration characteristics of the initial distribution efficiency of each province, showing the differences in each province's initial distribution efficiency. The initial distribution efficiency for most provinces were inefficient, and only two provinces, Zhejiang and Yunnan, had an efficiency of 1, reaching the DEA frontier. Because Zhejiang is a frontier region for efficient energy production and Yunnan is an environmentally sound region, during the first carbon emission rights allocation scheme, these two provinces achieved high levels of energy efficiency. Although the efficiency of Jiangsu province had not reached the effective frontier, it had risen above 0.8, implying that there was still room

for growth. The remaining provinces had efficiencies ranging from 0.3 to 0.8, implying that the initial allocation of carbon emission rights to the power sector was inefficient in these provinces. Among them, Jiangxi had the lowest efficiency value, which was lower than 0.5, followed by Chongqing, which was lower than 0.7. We should concentrate our efforts in these two areas on conserving energy and lowering emissions in the future. In addition, the efficiency of carbon emission rights allocation showed a downward trend in most provinces over time (see Table 1), indicating that allocating carbon emission quotas using the historical method not only made the allocation efficiency low, but also decreased the allocation efficiency over time. As a result, carbon quotas must be recalculated to achieve maximum efficiency.

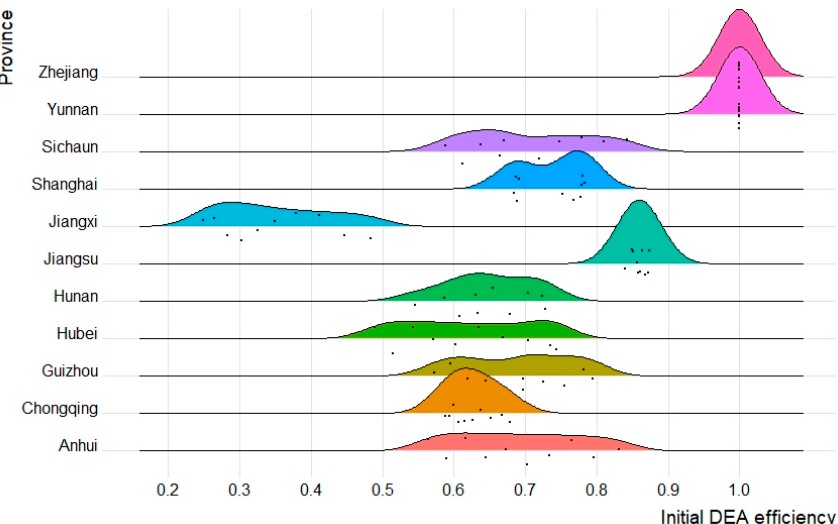

**Figure 1.** The initial carbon emission rights efficiency of the power sector in the provinces in the YREB from 2021 to 2030.

According to their geographical locations, the YREB can be divided into three regions: upper, middle, and lower reaches (Xing et al.) [41]. The lower reaches consist of Shanghai, Zhejiang, Jiangsu, and Anhui; the middle reaches consist of Hubei, Hunan and Jiangxi provinces; the upper reaches are made up of Chongqing, Sichuan, Guizhou, and Yunnan. Figure 2 depicts the trend of initial carbon emission allocation efficiency in the YREB's upper, middle, and lower reaches from 2021 to 2030. Overall, the initial carbon emission rights distribution efficiency in these three regions showed a downward trend, and there were significant differences. The YREB's downstream had the highest power efficiency, followed by the upstream, and the efficiency in the middle reaches was the lowest. This is because that most downstream provinces have advanced economic development and the power industry's technology is relatively advanced. These provinces were early adopters of new energy power generation technology, laying a good foundation for low-carbon development. The high efficiency of the upstream provinces lies in their good ecological environment. At the same time, with the strong support of national policies, the low-carbon economy of these provinces has been well developed. The provinces in the Yangtze River's middle reaches have developed power generation technology late, and have accumulated more carbon emissions, limiting the low-carbon development of these provinces.

The results of the research into the efficiency of initial allocation rights to carbon emissions show that DEA efficiency cannot be achieved with an initial allocation based on historical emissions, so the initial allocation must be adjusted iteratively employing the ZSG–DEA model. We scaled the allocation using Equation (2) until each province had an efficiency value close to 1, and the total carbon emissions remained constant as each iteration progresses. The basic principle was to keep total carbon emissions constant, adjust initial carbon emissions correctly, keep other input-output variables constant, and iterate continuously until the carbon emission allocation efficiency approaches 1. We only show

the adjustment process in 2021 due to space constraints, see Table 5. The carbon emission rights for the provincial power sector changed dramatically in the first two iterations, and the carbon emission adjustments varied widely across provinces. However, in the third iteration process, the adjustment amount of each province's carbon emission rights for the power sector were typically zero, which means that the final carbon emission quotas for the power sector in each province gradually tended to stabilise as the iteration progressed.

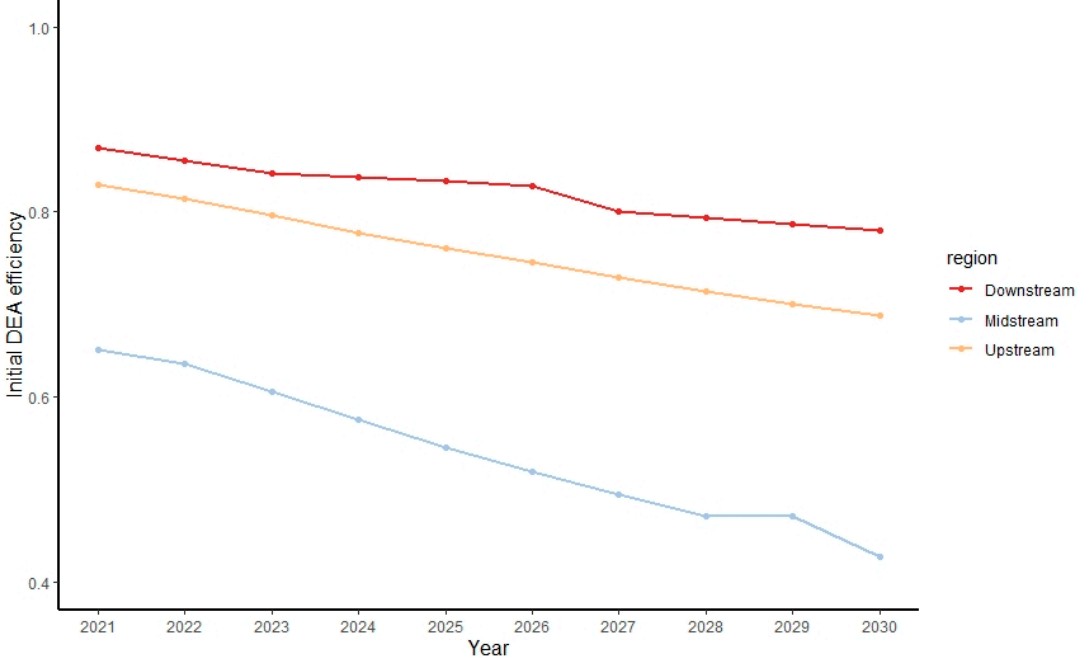

**Figure 2.** The initial carbon emission rights efficiency of the power sector in the upper, middle, and lower reaches of the YREB from 2021 to 2030.

**Table 5.** Optimization of allocation efficiency of carbon emission rights of power sector in the YREB in 2021.

| Province | Initial Quota (Million Tons) | Initial Efficiency | The First Iteration Value | First Iteration DEA Score | The Second Iteration Value | Second Iteration DEA Score | Final Quota (Million Tons) | Final Efficiency |
|---|---|---|---|---|---|---|---|---|
| Shanghai | 48.0822 | 0.7829 | 45.9293 | 0.8235 | 39.9394 | 0.9881 | 39.5591 | 0.9998 |
| Jiangsu | 176.7200 | 0.8476 | 174.6755 | 0.9314 | 169.1400 | 0.9966 | 168.9275 | 1.0000 |
| Zhejiang | 127.2981 | 1.0000 | 151.1066 | 1.0000 | 159.7891 | 1.0000 | 160.2785 | 1.0000 |
| Anhui | 65.4445 | 0.8317 | 65. 6307 | 0.9647 | 66.8642 | 0.9973 | 66.8731 | 0.9999 |
| Jiangxi | 38.3796 | 0.4840 | 24.6997 | 0.8726 | 22.8662 | 0.9927 | 22.7640 | 1.0000 |
| Hubei | 58.7681 | 0.7435 | 53.4204 | 0.9186 | 51.8097 | 0.9957 | 51.7297 | 1.0000 |
| Hunan | 50.9007 | 0.7276 | 45.5586 | 0.9374 | 45.1416 | 0.9966 | 45. 1183 | 1.0000 |
| Chongqing | 30.4770 | 0.6788 | 25.9780 | 0.9301 | 25.5896 | 0.9961 | 25.5648 | 1.0000 |
| Sichuan | 71.1966 | 0.8422 | 72.1122 | 0.9738 | 74.1675 | 0.9986 | 74.2807 | 1.0000 |
| Guizhou | 41.4484 | 0.7942 | 40. 1791 | 0.9542 | 40.5458 | 0.9975 | 40.5637 | 1.0000 |
| Yunnan | 50.3941 | 1.0000 | 59.8190 | 1.0000 | 63.2562 | 1.0000 | 63.4499 | 1.0000 |

Figure 3 shows the efficiency after each iteration. It is clear that more provinces were approaching the DEA frontier as the iterative process continued. In particular, only two provinces, Zhejiang and Yunnan reached the production frontier in the initial distribution.

In the first iteration, the distribution efficiency of most provinces exceeded 0.9, and after three iterations, almost all provinces had reached the production frontier. All the efficiency values were 1, indicating that all provinces' reallocated carbon allowances were nearly optimal after the third iteration.

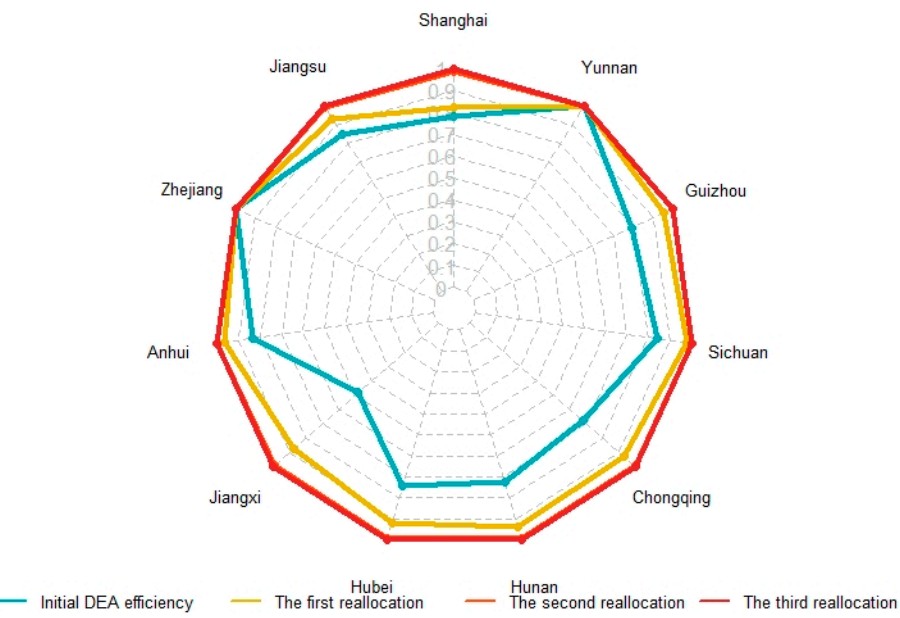

**Figure 3.** Distribution efficiency of the power sector in each province after each iteration in 2021.

Figure 4 depicts the adjusted amount of carbon emission allowances after three iterations. Carbon emission quotas must be increased in some provinces, while others must decrease quotas to achieve maximum efficiency. Provinces with higher initial efficiency had the largest increases in carbon emission quotas, including Zhejiang and Yunnan. Carbon emission allowances must be lowered in most provinces, with Jiangxi experiencing the greatest reduction. Because the adjustment equals zero, high-performing provinces should receive more carbon allowances from other provinces, whereas less efficient provinces should further reduce their carbon allowances, which means tightening $CO_2$ controls and setting higher emission reduction targets.

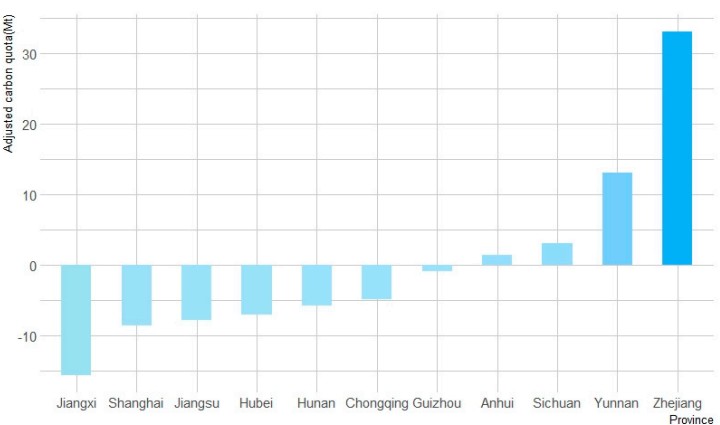

**Figure 4.** Quota adjustment of the power sector in each province after each iteration in 2021.

To clarify the power sector's pressure to reduce emissions in the provinces of the YREB, we used the iterative amount to achieve the optimal efficiency as the carbon emission quota that each province needs to be assigned. The amount of carbon emission quota means the carbon emission reduction pressure faced by each province. Four representative years were chosen, 2021, 2024, 2027, and 2030, and the spatial distribution of carbon emission

allowances in the YREB's power industry from 2021 to 2030 was obtained using the ArcGIS 10.2, as shown in Figure 5. Power carbon emission quotas are colour-coded and divided into six levels ranging from low to high. Overall, there were significant differences in power industry's carbon emission quotas across provinces, and they showed different spatial characteristics over time. During the previous period (2021–2027), the proportion of carbon emission quotas in Yunnan, Guizhou, and Hunan increased over time; the later period (2027–2030) was relatively balanced, and each province's carbon emission quotas reached a relatively stable state.

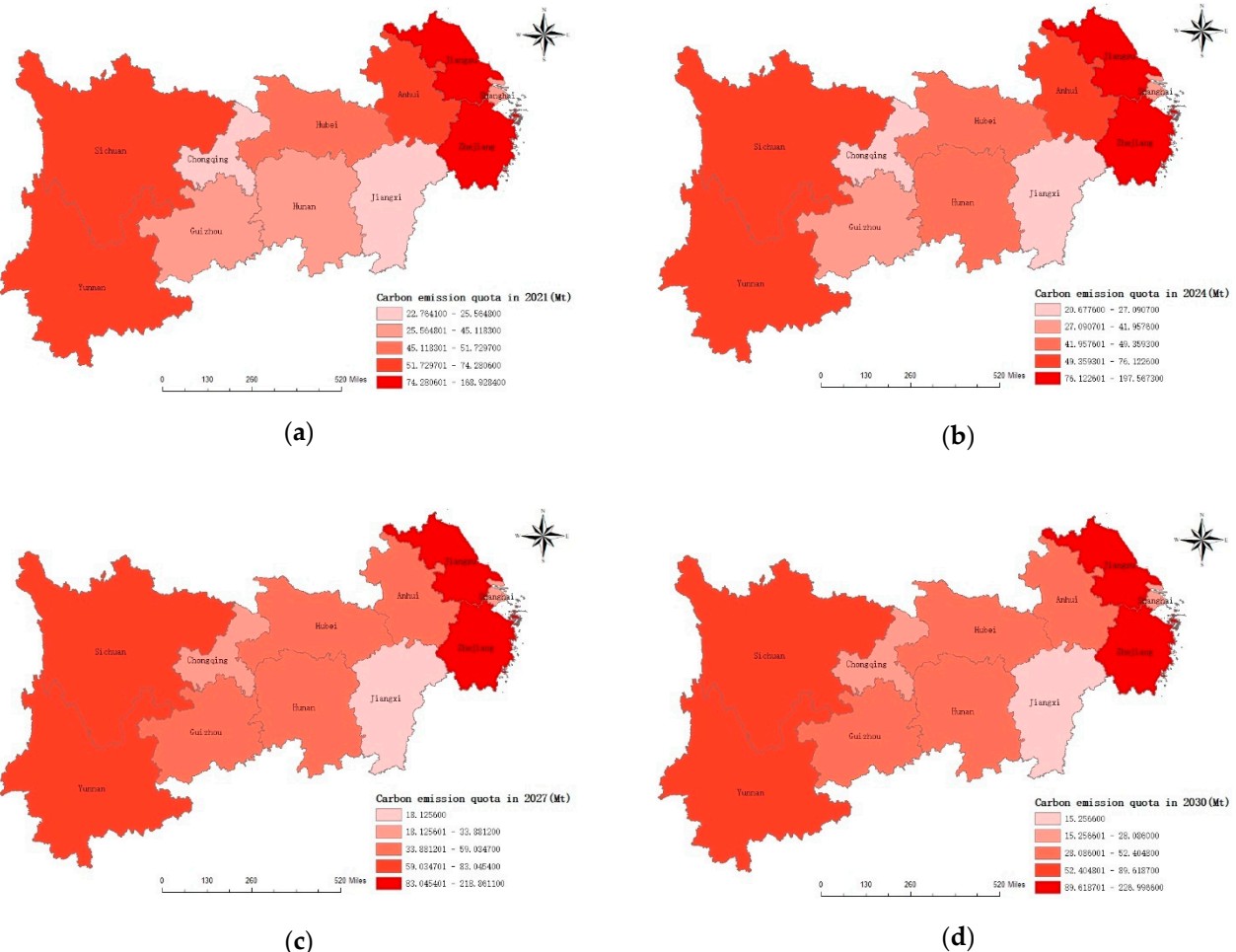

**Figure 5.** Spatial distribution of carbon emission quotas in the YREB's power sector. (**a**) Carbon emission quota of the power sector in each province in 2021 (Mt); (**b**) Carbon emission quota of the power sector in each province in 2024 (Mt); (**c**) Carbon emission quota of the power sector in each province in 2027 (Mt); (**d**) Carbon emission quota of the power sector in each province in 2030 (Mt).

Specifically, most provinces in the YREB's upper and lower reaches were distributed more carbon emission rights, and most provinces in the middle reaches were distributed less. This means that the power sector in the middle reaches of the province faces a greater challenge in emission reduction, because of the large number of thermal power generation tasks in the YREB's middle reaches and the excessive accumulation of carbon emissions.

Specific to each province, Zhejiang and Jiangsu were distributed the highest carbon emission quotas in these four times. On the one hand, Zhejiang and Jiangsu have strong emission reduction capabilities due to the economy's rapid growth and the use of technologies for advanced power generation. On the other hand, their economic development is not dependent on energy supply because of their relatively well-developed industrial structure and primarily high-tech industries. Following completion of their own emission-reduction

tasks, the two provinces can sell excess carbon emission rights to provinces with low-carbon emission rights, as well as provide technical assistance and financial subsidies. Followed by Zhejiang and Jiangsu, Sichuan and Yunnan were also allocated high carbon emission quotas. Their economies are not as developed, but their environment is better. Furthermore, the proportion of thermal power generation in Sichuan and Yunnan is small, as is the burden of carbon emissions. Jiangxi and Chongqing were allocated the fewest carbon emission quotas. These two provinces are under greater pressure to reduce emissions and should implement a variety of emission-control measures. The main feature of energy consumption structure in Jiangxi is coal-based, which is the main reason for increasing the pressure on Jiangxi to reduce emissions. At the same time, the power generation technology adopted in Jiangxi is still relatively backward, and it is strongly dependent on high-energy energy. Chongqing's pressure to reduce emissions stems primarily from its high-emission industrial structure. The industrial economy is the backbone of Chongqing's development as a city dominated by heavy industry, but it also brings high energy consumption and emissions. These two provinces should take stronger emission reduction measures, including optimization of energy structure and industrial structure, as well as technological innovation. The remaining provinces were allocated carbon emission quotas that were relatively balanced and low. This means they face stricter carbon emission restrictions. To meet their 2030 emission reduction targets, these provinces should implement a variety of emission-cutting measures. Based on the above analysis, Zhejiang, Jiangsu, Sichuan and Yunnan are more likely to be sellers in the carbon emissions trading market, while Jiangxi and Chongqing are more likely to be buyers in the carbon emissions trading mar

## 5. Conclusions and Policy Recommendations

In this paper, the carbon emission allocation efficiency of the power industry in 11 provinces of the YREB was calculated using the ZSG–DEA model from 2021 to 2030. Each province's carbon emissions quotas were redistributed to achieve maximum efficiency. The results show that:

(1) The power sector's initial distribution efficiency was inefficient in most provinces, only Zhejiang and Yunnan had reached the production frontier. Jiangxi had the lowest distribution efficiency, which was less than 0.5, and Chongqing's efficiency was less than 0.7. In these two provinces, we should concentrate on energy conservation and emission reduction. The efficiency of carbon emission rights allocation in most provinces showed a downward trend from 2021 to 2030, indicating that allocating carbon emission quotas using the historical method not only reduced allocation efficiency, but also decreased over time.

(2) The initial distribution efficiency of the power sector varied greatly between the YREB's upper, middle, and lower reaches. The downstream region had the highest power efficiency, followed by the upstream, and the middle region had the lowest. Because of the redistribution of carbon emission rights, most provinces in the upper and lower reaches had more carbon emission rights than middle reaches, implying that the power sector in the middle reaches faces greater emission reduction challenges.

(3) The carbon emission rights of the power industry varied greatly across provinces, and it exhibited different spatial characteristics over time. The early stage's carbon emission quota (2021–2027) varied greatly, while the later stage (2027–2030) was relatively balanced. Carbon emission quotas were higher in Zhejiang, Jiangsu, Sichuan, and Yunnan, which are more likely to become carbon trading market sellers. Jiangxi and Chongqing had lower carbon emission quotas and are thus more likely to participate in the carbon emission trading market as buyers. Other provinces' carbon emission quotas were relatively balanced.

This paper makes the following policy recommendations based on the research findings to help the YREB's power sector meet the 2030 emission reduction target.

(1) Differentiated carbon emission reduction goals and strategies must be established by local governments. Different targets should be set based on the different emission reduction pressures in each province. Provinces with a developed economy and a reasonable energy structure can set loose emission reduction targets, whereas provinces with a

developing economy and a heavy reliance on energy should set strict emission reduction targets. Furthermore, provincial power departments should fully exploit their own resource advantages and develop their own emission reduction strategies.

(2) The government should increase its investment in science and technology, as well as speed up the development of new power generation technologies such as clean energy and a reduction in thermal power generation. Provincial power departments should be encouraged to implement appropriate advanced power generation technologies, and the government can provide technical assistance to some provinces with weak economies and slow technological development.

(3) The industrial structure should be adjusted to speed up industrial upgrading. The unreasonable industrial structure seriously restricts the low-carbon development of various provinces. Further adjustments and optimization of the industrial structure are required, as well as the avoidance of energy-intensive industries and the active encouragement of the development of strategic emerging industries.

(4) The government can create fiscal policies to lower technology-related costs and provide financial assistance to provinces that are having difficulty reducing emissions.

(5) Regional cooperation should be strengthened, and carbon emission trading market should be improved. Most provinces are eligible to join the carbon emissions trading market. Through formulating reasonable market policies and regulating carbon emissions quota trading effectively, it is possible to achieve a coordinated economic and environmental development.

**Author Contributions:** Conceptualization, methodology, software, investigation, writing—original draft, writing—review and editing, project administration, and funding acquisition (D.M.); conceptualization, methodology, data curation, visualization, funding acquisition (Y.X.); validation, writing—review and editing, and funding acquisition (N.Z.). All authors have read and agreed to the published version of the manuscript.

**Funding:** This research was funded by [the Project of Chongqing Social Science Planning Project of China] grant number (No. 2020QNGL38) and The APC was funded by [the Humanities and Social Sciences Research Program of Chongqing Education Commission of China] grant number (No. 20SKGH169).

**Institutional Review Board Statement:** Not applicable.

**Informed Consent Statement:** Not applicable.

**Data Availability Statement:** All the data can be found in http://www.stats.gov.cn/tjsj/ndsj/ (accessed on 15 March 2022).

**Acknowledgments:** We are particularly grateful for the financial support from the Project of Chongqing Social Science Planning Project of China (No. 2020QNGL38), Humanities and Social Sciences Research Program of Chongqing Education Commission of China (No. 20SKGH169), and we are grateful to anonymous reviewers and editors for their comments and suggestions on this paper.

**Conflicts of Interest:** The authors declare no conflict of interest.

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
