# Peer review of "Optimization and Spatiotemporal Differentiation of Carbon Emission Rights Allocation in the Power Industry in the Yangtze River Economic Belt"

_sustainability, doi:10.3390/su14095201_

Round 1
Reviewer 1 Report
The authors have done an interesting job, however in my opinion the clarity of the article can be improved.
Numerical calculations may be correctly carried out, but the information that establishes their values ​​is not clearly evidenced.
In general, they must explain the current rights distribution criteria and the reason why they are efficient-inefficient.
The assignment of the initial factors and the redistribution criteria in the successive iterations is not clear.
Related to this point, on page 3 line 122 they indicate that they ignore other more efficient distribution principles. Do they have something to do with the generation technologies used?
On page 4, line 179, they indicate that global emissions are the same, it is a limiting factor of the total electrical energy generated or it is assumed that the increase in efficiency also assumes the increases in consumption derived from economic growth.
The inclusion of a table of abbreviations would facilitate understanding, and even describe them before citing them, as an example ZSG-DEA are cited several times and are described on page 3, lines 122 and 127.
References should be checked, some are missing as in:
- page 2 line 49 China´s current power ...
- page 3 line 100, the Gini coefficient
- table 2 and 3, source of the coefficients
- page 9 line 321, missing in references
etc..
Reviewer 2 Report
It is acknowledged this paper continues with a well-worn and important theme of managing and balancing China’s electrical power production/consumption. Following that route, this paper can serve some use to decisionmakers and perhaps even challenge the vested interests to make a more robust case for what they want.
I state at the outset that I do not put much trust in the DEA approach of the 1978 Charnes, Cooper, Rhodes paper which was a revision of the Farrell Efficiency Measure. I assume the units of the DMU are those set out in Table 1. How the units are quantified is not clear to me. Their injections into the formulae with coefficient weightings I accept as being justified because it would take a new and independent team a long time to follow and challenge each step.
I do not see where the quotas are presented except in Figure 5. What do the numbers in the code represent. Is the last number for 2030, “226.996600” for millions of tons? Of carbon or CO2?
I accept this paper fits a pattern that is used in discussions and though I suggest it can be improved I do not think the authors would agree with my critique. There is nothing wrong with the text, spelling, grammar, punctuation. In my opinion it can be published and will be read by some in the quota decision-making processes
